# A Pilot Study on the Inter-Operator Reproducibility of a Wireless Sensors-Based System for Quantifying Gait Asymmetries in Horses

**DOI:** 10.3390/s22239533

**Published:** 2022-12-06

**Authors:** Iris Timmerman, Claire Macaire, Sandrine Hanne-Poujade, Lélia Bertoni, Pauline Martin, Frédéric Marin, Henry Chateau

**Affiliations:** 1Ecole Nationale Vétérinaire d’Alfort, USC INRAE-ENVA 957 BPLC, CIRALE, 94700 Maisons-Alfort, France; 2LIM France, Labcom LIM-ENVA, 24300 Nontron, France; 3Laboratoire de BioMécanique et BioIngénierie (UMR CNRS 7338), Centre of Excellence for Human and Animal Movement Biomechanics (CoEMoB), Université de Technologie de Compiègne (UTC), Alliance Sorbonne Université, 60200 Compiègne, France

**Keywords:** horse, locomotion, gait analysis, repeatability, reproducibility, inertial measurement units

## Abstract

Repeatability and reproducibility of any measuring system must be evaluated to assess possible limitations for its use. The objective of this study was to establish the repeatability and the inter-operator reproducibility of a sensors-based system (EQUISYM^®^) for quantifying gait asymmetries in horses.. Seven wireless IMUs were placed on the head, the withers, the pelvis, and the 4 cannon bones on three horses, by four different operators, four times on each horse, which led to a total of 48 repetitions randomly assigned. Data were collected along three consecutive days and analysed to calculate total variance, standard deviation and the variance attributable to the operator on multiple asymmetry variables. Maximal percentage of variance due to the operator (calculated out of the total variance) was 5.3% and was related to the sensor placed on the head. The results suggest a good reproducibility of IMU-based gait analysis systems for different operators repositioning the system and repeating the same measurements at a succession of time intervals. Future studies will be useful to confirm that inter-operator reproducibility remains valid in larger groups and on horses with different degrees of locomotor asymmetry.

## 1. Introduction

Locomotor diseases associated with lameness are among the most frequent disorders affecting horses. Hence, they represent a major part of the working time and incomes of equine veterinarians [1,2]. Even for the most experienced veterinarians, the evaluation of lameness remains subjective and the diagnosis complicated [3,4,5]. Subjectivity increases when the lameness is slight [6], and some bias in the evaluation makes the repeatability of the visual examination questionable, as with the expectation of improvement after nerve block [7].

The will to move towards a more objective evaluation allowed the rise of several tools for gait analysis. Stationary force plate and Motion Capture are the most precise and accurate methods, but they cannot be used in current practice, in the field [8]. The needs of veterinary practitioners are for a portable, trustworthy, fast and easy tool to use in the field. Inertial Measurement Units (IMUs) appear then as the most suitable alternative [9].

For instance, EQUISYM^®^ is a tool designed to assist veterinarians in the daily clinical diagnosis of locomotor disorders in horses. It provides quantified locomotion data and guides the diagnosis of lameness with objective indicators based on the calculation of asymmetries between left and right limbs during locomotion at the trot. In this context, the system can be used by different veterinarians at different times. In order to be able to compare results with each other, to ensure long-term locomotor monitoring or to carry out multi-centre studies, it is necessary to verify that the system is capable of giving an equivalent measurement regardless of the operator using it, including novices.

For this purpose, the repeatability and reproducibility of the system, among other factors, must be assessed. The repeatability condition of measurement is defined as “condition of measurement, out of a set of conditions that includes the same measurement procedure, same operators, same measuring system, same operating conditions and same location, and replicate measurements on the same or similar objects over a short period of time” and the reproducibility condition of measurement as “condition of measurement, out of a set of conditions that includes different locations, operators, measuring systems, and replicate measurements on the same or similar objects” [10].

IMUs-based systems for quantifying lameness in horses have been assessed by showing the correlation or the concordance with Motion Capture systems used as a gold standard [11,12,13,14,15]. However, few studies aimed to assay the repeatability of the measurements. For instance, Keegan and al. studied an IMUs-based system repeatability in an uncontrolled environment on 236 horses [16]. After instrumentation of each horse, they were trotted twice at five minutes’ intervals in a straight line, in various places and surface conditions. Another study aimed to determine repeatability of gait variables such as the symmetry of several segments (metacarpal and metatarsal region, hind limb asymmetry…) and joint range of motion in the sagittal and coronal planes, during trotting under a controlled treadmill exercise [17]. In this study, ten non-lame horses were instrumented and then trotted three times a day, three days a week, over a three-week period. The repeatability was considered as high for most of the variables. A third study evaluated the influence of biological variations over time by trotting fourteen thoroughbred horses in a straight line once a day for five consecutive days, and then once a week for five consecutive weeks while horses were in race training [18]. Daily and weekly repeatability was lower than in the previous study [17]. Mainly, horses were of varying training and movement asymmetry levels and it was not possible to dissociate the effects of the sensors placement from the biological variability. None of these studies have evaluated the inter-operator reproducibility. To the best of our knowledge, it has not been studied yet.

In this context, the goal of our study was to focus on the effect of IMUs placement by different operators in order to assess reproducibility and to verify that the system can be used reliably regardless of the user’s level of expertise, under all the conditions of the locomotor examination (straight line and circles).

The objective of our study was then to evaluate the repeatability and the inter-operator reproducibility of a gait analysis system, EQUISYM^®^, in a standardised environment, on a straight line and in circles, with different users. It was hypothesised that the variance attributed to the operator is negligible compared to the total variance.

## 2. Materials and Methods

### 2.1. Horses

Three trotters from the Centre d’Imagerie et de Recherche sur les Affections Locomotrices Equines (CIRALE) were included: a 13 years old gelding, a 14 years old mare and a 27 years old gelding. These three horses were not on any training beyond the experiment and no subjective gait change was noticed over the three days of experiment.

### 2.2. Instrumentation

Horses were equipped with EQUISYM^®^, composed of 7 wireless IMUs (tri-axial accelerometer ±16 × gravity, tri-axial gyroscope ±2000 deg/s) placed on the head, the withers, the pelvis, and the 4 cannon bones (Figure 1). They recorded at 200 Hz during a minimum of 20 trot strides on straight line and in circles.

### 2.3. Protocol

The facilities were composed of a straight line of 30 m and a circle of 14 m in diameter. Four different operators were selected: two of them had never placed the sensors before, whereas the other two operators were used to doing it. A demonstration was given for the operators who had never placed it before the experiment. Each operator placed the sensors four times on each horse along three consecutive days, which led to a total of 48 repetitions randomly assigned. To avoid the habituation effect, each operator waited at least one hour between two consecutive positionings. When the measurement system was placed, the horse was trotted in-hand on a hard surface, firstly in a left rein circle, then in a right rein circle and finally in a straight line. The protocol was intended to mimic the standardised conditions of a locomotor examination. The mean +/− standard deviation (SD) of the recorded strides was distributed as following: 29.5 +/− 5.5 strides on straight line, 32.1 +/− 9.5 on right rein circle, 36.5 +/− 9.4 on left rein circle.

The handler was the same person during the three days of the experiment. Each horse performed a maximum of three successive repetitions in order to limit the influence of fatigue. Between two sessions of three repetitions, each horse had a minimum break of one hour.

### 2.4. Data Processing

Data were analysed using custom-written Matlab2020a (The MathWorks, Natick, MA, USA) scripts. Stance sequences were selected based on sequences of the cannon bones gyroscopic signal. Foot-on and foot-off were determined from cannon bones IMUs’ signals thanks to the method developed by Hatrisse and al. [19]. The dorso-ventral acceleration signal was integrated twice to obtain displacement values, and was high-pass filtered using a fourth-order Butterworth filter with a cut-off frequency set to 1 Hz. The choice of this cut-off frequency was made after checking that the high-pass filter suppresses the slow drift but does not distort the shape of the signal, following the recommendations of Serra Bragança et al. [20]. Vertical displacements of the head, withers and pelvis were segmented into strides. One stride was defined as the time between two consecutive foot-on of the left forelimb.

From the vertical displacement of the head (-H), withers (-W) and pelvis (-P) occurring along a stride, 4 variables were calculated for each sensor location. The following Asymmetry Indexes (AI), expressed as percentage of the maximal range of motion within a stride, were used to compare left vs. right part of the stride [21]: AI-Min was the left-right difference of the lowest altitude point; AI-Max was the left-right difference of the highest altitude point, AI-up was the left-right difference of the upward range of motion during the propulsion phase, AI-down was the left-right difference of the downward range of motion during the damping phase, and AI-stance-duration was the left-right difference of the duration of the stance phase. Positive AI value indicated a weaker movement amplitude or a shorter stance phase duration (AI-stance-duration) during the right stance than during the left stance, and negative AI value indicated the opposite.

### 2.5. Statistical Analysis

To evaluate the repeatability and the inter-operator reproducibility, for each AI presented above and each condition (i.e., straight line, left and right rein circle), the total variance, as well as the corresponding SD, and the variance attributable to the operator were calculated. To do this, a model of analysis of variance was computed with operators and horses as random effects [22]. As the rank of repetition and the experience of the operator could also be a source of variation, the model was adjusted on these two factors as fixed effects. The statistical analysis was performed with the RStudio software 4.0.3 (RStudio Inc., Boston, MA, USA).

## 3. Results

Straight line results are shown in Table 1 and Appendix A. In a straight line, the total variance was the highest for the index AI-Up-H (723.3) and the lowest for AI-Stance-Duration-HIND (11.1). Total variance and SD were higher for head-related AIs than for the withers or pelvis. The SD ranged between 18.6% and 26.9% for head-related AIs, and between 3.3% and 14.2% for pelvis and withers-related AIs. For all indexes, the percentage of variance attributable to the operator was close to 0, with a maximum value of 2.1% for AI-Stance-Duration-FRONT.

Left rein circle results and right rein circle results are presented in Table 2 and Appendix A. In both left and right rein circles, total variance was maximum for AI-Up-H (respectively 1448.2 and 2089.4), and minimum for AI-Stance-Duration-FRONT (respectively 18.1 and 36.7). As in a straight line, the total variance and SD were higher for head-related AIs. The SD varied in a similar range of values, respectively between 15.0% and 38.1% in the left rein circle and between 13.5% and 45.7% in the right rein circle for head-related AIs, and between 4.2% and 26.2% in the left rein circle and between 6.3% and 31.3% in the right rein circle for pelvis and withers-related AIs. AI-Down was systematically associated with a larger variance than the other AIs calculated for the withers and the pelvis.

Similar to the straight line, the variance attributable to the operator in the circle was close to zero for most of the AIs and reached a maximum of 4.6% of the total variance for AI-Stance-Duration-HIND in the left rein circle, and a maximum of 5.3% of the total variance for AI-Max-H in the right rein circle.

## 4. Discussion

The purpose of this study was to evaluate repeatability and reproducibility between operators of an IMU-based gait analysis system used to quantify asymmetry indexes in horses trotting in a straight line and circles under the same circumstances as a standard locomotor clinical exam. In our study, the variance attributable to the operator was very low, which tends to validate our starting hypothesis: the asymmetry indexes are repeatable when one or several operators set up the sensors several times, when the horse trots on a hard surface, in a straight line or in a circle.

In this study, the variance attributable to the operators was slightly higher on a right rein circle, for head-related indexes (up to 5.3% of total variance). No obvious explanation can be attributed to this observation except than that the effect of a small misplacement of the sensors by the operator may have been slightly higher on the circle than for the straight line.

As slight biological variations may be clinically relevant and many operators may be required to use the system, including those with little training, assessing reproducibility is essential to enable comparison of results obtained at different times, by different operators. Our results show that on the 42 asymmetry indexes measured (14 in a straight line, 14 in a left hand circle and 14 in a right hand circle), the variance attributable to the operator counted for less than 1% of the total variance in 81% of cases. For each exercise, the head-related indexes showed the highest SD (between 13.5% and 45.7%). In contrast, the withers-related variables showed the lowest SD (between 4.2% and 23.4%). The index AI-Down was associated with a larger variance than the other AIs for the withers and for the pelvis, indicating a higher variability of this index and thus a more complex interpretation.

Observed values for SD for the straight line were close to the values calculated from a group of 49 sound horses with the same system [21]. SD values for the left and right rein circles were slightly higher. It has been shown that moving in circles induces a movement asymmetry which depends on speed and circle radius [23]. In our experiment none of these factors were strictly controlled, knowing that we wanted to remain in the same conditions as the standard clinical examination. This could explain the increased SD in circles, with a higher level of freedom, enabling the horse to change its body direction and speed [23,24]. It is also known that asymmetry on a circle is physiologically higher than in a straight line.

Total variance was higher for head related indexes than for the withers and pelvis, for the straight line and circles. Similar results have been found in previous experiments, in which head variability was higher than pelvis and withers variability [18,24]. This is probably due to a wider degree of freedom for the head, resulting in multiple head movements caused by external stimuli such as other horses passing by, or noises. These uncontrolled head movements during some of our experiments are likely to explain the increase in total variance for indexes related to the head. This result supports the use of a sensor on the withers, which presents a much lower variability when exploring forelimb lameness. 

After adjustment of our statistical model on the level of expertise of the operator and on the rank of repetition, nothing indicates that one of them had a significant effect. Hence, no confounding could have been made between the level of expertise of the operator and the variability induced by the operators.

The area of the experiment was fixed in our study, and our hypotheses were tested only on a hard surface. A similar study should be done on a soft surface to extrapolate our findings. The reproducibility should also be studied in different places. Furthermore, asymmetry of the included horses has been considered steady during the three days of the experiment but it is obvious that the total variance measured depends not only on the measurement system but also on the inevitable biological variation of the locomotor asymmetry, whether physiological or pathological.

To date, no study had been conducted to measure the inter-operator reproducibility of IMU’s to evaluate gait asymmetry in horses. Our study provides initial evidence that this type of system can be used by different operators, including novices, without major disruption to the measurement collection. It also confirms that the variability of the measurements is higher for the head-related indexes than for the withers and pelvis related indexes. The sample size (number of horses and operators) remains a limitation of this study. Ideally, future studies should be conducted to verify that inter-operator reproducibility remains valid in larger groups and on horses with different degrees of locomotor asymmetry.

## Figures and Tables

**Figure 1 sensors-22-09533-f001:**
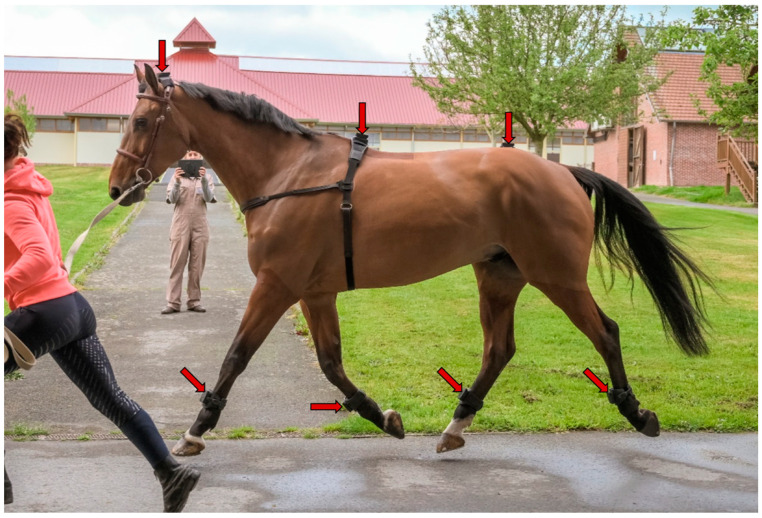
Horse equipped with EQUISYM^®^ system, composed by sensors placed on the head, the withers, the pelvis and the 4 cannon bones (indicated by red arrows).

**Table 1 sensors-22-09533-t001:** Total variance, total standard deviation and percentage of variance due to the operators (calculated out of the total variance) for four asymmetry indexes (AI -Up, -Down, -Max and -Min) calculated on the head (H), the withers (W) and the pelvis (P) for 48 repeated measurements (4 operators, 3 horses, 4 repetitions) performed on a straight line.

**AI**	**Total Variance**	**Total Standard Deviation**	**Percentage of Variance Due to the Operators (%)**
AI-Up-H (%)	723.3	26.9	0.0
AI-Down-H (%)	683.0	26.1	0.0
AI-Max-H (%)	346.1	18.6	0.5
AI-Min-H (%)	506.3	22.5	0.0
AI-Up-W (%)	35.6	6.0	0.7
AI-Down-W (%)	119.0	10.9	0.0
AI-Max-W (%)	53.8	7.3	1.1
AI-Min-W (%)	39.9	6.3	0.0
AI-Stance-Duration-FRONT (%)	57.4	7.6	2.1
AI-Up-P (%)	97.4	9.9	0.6
AI-Down-P (%)	202.4	14.2	0.0
AI-Max-P (%)	89.2	9.4	0.4
AI-Min-P (%)	90.0	9.5	0.1
AI-Stance-Duration-HIND (%)	11.1	3.3	0.0

**Table 2 sensors-22-09533-t002:** Total variance, total standard deviation and percentage of variance due to the operators (calculated out of the total) for four asymmetry indexes (AI -Up, -Down, -Max and -Min) calculated on the head (H), the withers (W) and the pelvis (P) for 48 repeated measurements (4 operators, 3 horses, 4 repetitions) performed on right and left rein circle.

	**Left Rein Circle**			**Right Rein Circle**		
**AI**	**Total Variance**	**Total Standard Deviation**	**Percentage of Variance Due to the Operators (%)**	**Total Variance**	**Total Standard Deviation**	**Percentage of Variance Due to the Operators (%)**
AI-Up-H (%)	1448.2	38.1	0.5	2089.4	45.7	4.2
AI-Down-H (%)	521.0	22.8	0.0	963.1	31.0	0.0
AI-Max-H (%)	224.2	15.0	0.5	182.6	13.5	5.3
AI-Min-H (%)	765.0	27.7	0.0	1314.9	36.3	2.8
AI-Up-W (%)	111.2	10.5	3.6	403.6	20.1	0.0
AI-Down-W (%)	179.6	13.4	0.0	513.7	22.7	1.2
AI-Max-W (%)	102.2	10.1	1.0	97.9	9.9	0.0
AI-Min-W (%)	69.0	8.3	0.0	548.8	23.4	0.0
AI-Stance-Duration-FRONT (%)	18.1	4.2	1.0	115.5	10.7	0.0
AI-Up-P (%)	254.4	15.9	0.0	981.6	31.3	0.0
AI-Down-P (%)	687.7	26.2	0.0	124.5	11.2	0.0
AI-Max-P (%)	174.7	13.2	0.0	226.5	15.0	0.0
AI-Min-P (%)	397.9	19.9	0.0	334.7	18.3	0.4
AI-Stance-Duration-HIND (%)	18.4	4.3	4.6	36.7	6.1	0.0

## Data Availability

The data that support the findings of this study are available from the corresponding author upon reasonable request.

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
