# Peer review of "A Pilot Study on the Inter-Operator Reproducibility of a Wireless Sensors-Based System for Quantifying Gait Asymmetries in Horses"

_sensors, 2022, doi:10.3390/s22239533_

Round 1

Reviewer 1 Report

Dear Authors,

In my opinion, interesting and valuable results have been presented in this manuscript.  My only concern regarding this manuscript is that the size of the measurement sample is on the lower boundaries.

In Line 109 “scripts).“ should be replaced with ") scripts.“

The 1 Hz cut-off frequency seams rather low, especially if higher harmonics are considered. Any argumentation on the choice?

Author Response

Dear Reviewer,

We would like to thank you for your comments and suggestions on our manuscript. All requests have been taken into account in the revision of this manuscript.

Please find below our point-by-point responses to your report.

Reviewer request: In my opinion, interesting and valuable results have been presented in this manuscript.  My only concern regarding this manuscript is that the size of the measurement sample is on the lower boundaries.

Authors’ answer: We are aware of the measurement sample size. Nevertheless, due to constraints on the experimental protocol (time, horses’ recruitment, availability of operators for several consecutive days of experiment and of course financial cost), we enrolled the most appropriate number of horses and operators as we can. To overcome this issue, we opted to increase the number of repetitions for each couple (operator x horse) to collect 48 samples (3 horses x 4 operators x 4 repetitions) to have an acceptable total sample size to perform analyses.

However, we are aware that the sample size is small in our case and recognise that our results need to be interpreted carefully and discussed in an informed way.

Therefore, in order to make this point clearer in the paper, we suggest to add the following sentence to the discussion:

"The sample size (number of horses and operators) could be considered as a limitation of this study. To overcome this issue, we have chosen to increase the number of repetitions for each couple (operator x horse) to collect 48 samples (3 horses x 4 operators x 4 repetitions) to have an acceptable total sample size to perform analyses.”

Reviewer request: In Line 109 “scripts).“ should be replaced with ") scripts.“

Authors’ answer: Thank you for your comment. The change has been done in the new version of the article.

Reviewer request: The 1 Hz cut-off frequency seams rather low, especially if higher harmonics are considered. Any argumentation on the choice?

Authors’ answer: We need to emphasize on the point that the filter used is a high-pass filter. All high frequency harmonics have been preserved as no low-pass filter has been used. The high-pass filter with a very low frequency cut-off only aims at rectifying the signal by eliminating the small low frequency drift linked to the integration of the signal. The elimination of this drift is made possible by the fact that the movement is cyclic and that it is therefore physiological that the arrival altitude is the same as the altitude of the departure.

We thank you again for your help and we hope that these answers and the new version of our manuscript will meet your expectations.

Best regards

Reviewer 2 Report

Line 112 and 113 you are stating “… high-pass filtered using a fourth-order Butterworth filter with a cut-off frequency set to 1 Hz.” Is this really what you were doing? So, you are discarding the low frequencies between 0 and 1 Hz, which contain a big portion of the useful signal but keeping the noise of the higher frequencies? In case you are using a low-pass filter with a 1 Hz cutoff you are missing an important part of the signal as well since the sampling theorem clearly indicates a higher cutoff is necessary.

Line 169 and 170: “In our study, the variance attributable to the operator was very low, …”. This is expectable since you showed in figure 1 very precise how to put the sensors. Even in the unprobeable case of the sensors being placed in a wrongly rotated position compared to the instructions, there exist always a transformation correcting the outcome.

The validity of measuring systems is of prime importance for all experiential work, which of cause applies to “… wireless sensors-based system for quantifying horse’s gait asymmetries …” as well. Validity should be mentioned in any empirical study. An analysis exclusively and only concerned with the validity of the measuring system within a specific application makes sense only, if no or not enough knowledge is available. The argument in the study is only few “… studies aimed to assay the repeatability …” of “… quantifying lameness in horses …” as an application mentioned. True! But there exist tons of papers showing the validity of IMU-based systems specifically applied to human movement. This journal is an excellent hub for this kind of studies, which are not mentioned in your paper at all. Bipedal walking most probably has a bigger chaotic ratio compared to four-legged walking, which would allow to use the well-known results of IMU-validity in studies on human movement to be considered. This study does not improve the knowledge on IMU validity, which validity is adequately shown already. A meaningful contribution to the scientific literature, however, would be given if e.g., the lameness in horses compared to uninjured horses with the specific mentioning of the signal to noise ratio would be given.

Author Response

Dear Reviewer,

We would like to thank you for your comments and suggestions on our manuscript. All requests have been taken into account in the revision of this manuscript.

Please find below our point-by-point responses to your report.

Reviewer request: Line 112 and 113 you are stating “… high-pass filtered using a fourth-order Butterworth filter with a cut-off frequency set to 1 Hz.” Is this really what you were doing? So, you are discarding the low frequencies between 0 and 1 Hz, which contain a big portion of the useful signal but keeping the noise of the higher frequencies? In case you are using a low-pass filter with a 1 Hz cutoff you are missing an important part of the signal as well since the sampling theorem clearly indicates a higher cutoff is necessary.

Authors’ answer: We need to emphasize on the point that the filter used is a high-pass filter. All high frequency harmonics have been preserved as no low-pass filter has been used. The high-pass filter with a very low frequency cut-off only aims at rectifying the signal by eliminating the small low frequency drift linked to the integration of the signal. The elimination of this drift is made possible by the fact that the movement is cyclic and that it is therefore physiological that the arrival altitude is the same as the altitude of the departure.

Reviewer request: Line 169 and 170: “In our study, the variance attributable to the operator was very low, …”. This is expectable since you showed in figure 1 very precise how to put the sensors. Even in the unprobeable case of the sensors being placed in a wrongly rotated position compared to the instructions, there exist always a transformation correcting the outcome.

Authors’ answer: Thank you for your comment. This result was indeed expected and we hoped that the equipment specially designed to minimise this risk of error was sufficiently well engineered to allow comparisons of measurements between operators. We however strongly believe that this requires a scientific validation in order to remove any doubt and to ensure that the system used by any operator produces comparable results. This will be essential if we wish to compare exams with each other or use the system for multi-centre studies.

Regarding the unprobeable case of the sensors being placed in a wrong position (e.g. two sensors are switched), the algorithm can detect this and the system will not produce results. Thus, the risk of producing bad data is avoided. Our objective here was only to estimate the variability related to small misplacements of the sensors.

Reviewer request: The validity of measuring systems is of prime importance for all experiential work, which of cause applies to “… wireless sensors-based system for quantifying horse’s gait asymmetries …” as well. Validity should be mentioned in any empirical study. An analysis exclusively and only concerned with the validity of the measuring system within a specific application makes sense only, if no or not enough knowledge is available. The argument in the study is only few “… studies aimed to assay the repeatability …” of “… quantifying lameness in horses …” as an application mentioned. True! But there exist tons of papers showing the validity of IMU-based systems specifically applied to human movement. This journal is an excellent hub for this kind of studies, which are not mentioned in your paper at all. Bipedal walking most probably has a bigger chaotic ratio compared to four-legged walking, which would allow to use the well-known results of IMU-validity in studies on human movement to be considered. This study does not improve the knowledge on IMU validity, which validity is adequately shown already. A meaningful contribution to the scientific literature, however, would be given if e.g., the lameness in horses compared to uninjured horses with the specific mentioning of the signal to noise ratio would be given.

Authors’ answer: We definitely agree with your comment. However, the purpose of our study was not to assess the validity of IMU-based system in general (which is already done of course) but specifically to test the reproducibility of a system used in horses in the context of assessing locomotor disorders in this species. Our aim was to ensure that, when different people use the system, we can have confidence in the comparability of the results.

We have tried to make this point clearer in the introduction to avoid any misunderstanding. The following paragraph has been added in the introduction section:

“For instance, EQUISYM® is a tool designed to assist veterinarians in the daily clinical diagnosis of locomotor disorders in horses. It provides quantified locomotion data and guide the diagnosis of lameness with objective indicators based on the calculation of asymmetries between left and right limbs during locomotion at the trot. In this context, the system can be used by different veterinarians at different times. In order to be able to compare results with each other, to ensure long-term locomotor monitoring or to carry out multi-centre studies, it is necessary to verify that the system is capable of giving an equivalent measurement regardless of the operator using it, including novices.”

We thank you again for your help and we hope that these answers and the new version of our manuscript will meet your expectations.

Best regards

Reviewer 3 Report

The paper presents an experimental study regarding the repeatability and inter-operator reproducibility of a wireless sensor-based system for quantifying horse’s gait asymmetries.

The study is well conducted, and the results are well presented, the experimental tests are well described, and the data interpretation is adequate. Can the authors provide further information regarding the use of their study in practice? How it will be used in future research or implemented?

Best regards

Author Response

Dear Reviewer,

We would like to thank you for your comments and suggestions on our manuscript. All requests have been taken into account in the revision of this manuscript.

Please find below our point-by-point responses to your report.

The paper presents an experimental study regarding the repeatability and inter-operator reproducibility of a wireless sensor-based system for quantifying horse’s gait asymmetries.

The study is well conducted, and the results are well presented, the experimental tests are well described, and the data interpretation is adequate.

Thank you for your comment.

Reviewer request: Can the authors provide further information regarding the use of their study in practice? How it will be used in future research or implemented?

Authors’ answer: Thank you for your suggestion. We have tried to better describe the use of the tool in practice and how it will be used in future work and research project. We have added the following paragraph to the introduction section:

“EQUISYM® is a tool designed to assist veterinarians in the daily clinical diagnosis of locomotor disorders in horses. It provides quantified locomotion data and guide the diagnosis of lameness with objective indicators based on the calculation of asymmetries between left and right limbs during locomotion at the trot. In this context, the system can be used by different veterinarians at different times. In order to be able to compare results with each other, to ensure long-term locomotor monitoring or to carry out multi-centre studies, it is necessary to verify that the system is capable of giving an equivalent measurement regardless of the operator using it, including novices.”

Current and future work includes establishing thresholds between healthy and pathological horses and establishing the physiological variability of locomotor asymmetry to enable early detection of subtle changes in locomotion that may be indicative of injury, for early diagnosis and prevention. The system will also be used as a tool to monitor and quantify the effects of treatments of locomotor disorders in multicentre studies.

We thank you again for your help and we hope that these answers and the new version of our manuscript will meet your expectations.

Best regards

Round 2

Reviewer 2 Report

Publisher's decision

Author Response

Dear Reviewer,

Your comment “Publisher’s decision” has been taken into account.

We have responded point by point to all the editor's requests to update this new version of our article.

We thank you again for your help.

Best regards